# VALUE FUNCTION SPACES: SKILL-CENTRIC STATE ABSTRACTIONS FOR LONG-HORIZON REASONING

**Dhruv Shah**$^{\gamma\beta}$**, Peng Xu**$^{\gamma}$**, Yao Lu**$^{\gamma}$**, Ted Xiao**$^{\gamma}$
**Alexander Toshev**$^{\gamma}$**, Sergey Levine**$^{\gamma\beta}$**, Brian Ichter**$^{\gamma}$

$^{\gamma}$Google Research, Robotics @ Google
$^{\beta}$Berkeley AI Research, UC Berkeley

## ABSTRACT

Reinforcement learning can train policies that effectively perform complex tasks. However for long-horizon tasks, the performance of these methods degrades with horizon, often necessitating reasoning over and chaining lower-level skills. Hierarchical reinforcement learning aims to enable this by providing a bank of low-level skills as action abstractions. Hierarchies can further improve on this by abstracting the space states as well. We posit that a suitable state abstraction should depend on the capabilities of the available lower-level policies. We propose Value Function Spaces: a simple approach that produces such a representation by using the value functions corresponding to each lower-level skill. These value functions capture the affordances of the scene, thus forming a representation that compactly abstracts task relevant information and robustly ignores distractors. Empirical evaluations for maze-solving and robotic manipulation tasks demonstrate that our approach improves long-horizon performance and enables better zero-shot generalization than alternative model-free and model-based methods.

## 1 INTRODUCTION

For an agent to perform complex tasks in realistic environments, it must be able to effectively reason over long horizons, and parse high-dimensional observations to infer the contents of a scene and its affordances. It can do so by constructing a compact representation that is robust to distractors and suitable for planning and control. Consider, for instance, a robot rearranging objects on a desk. To succcessfully solve the task, the robot must learn to sequence a series of simple skills, such as picking and placing objects and opening drawers, and interpret its observations to determine which skills are most appropriate. This requires the ability to understand the capabilities of these simpler skills, as well as the ability to plan to execute them in the correct order.

Hierarchical reinforcement learning (HRL) aims to enable this by leveraging *abstraction*, which simplifies the higher-level control or planning problem. Typically, this is taken to mean abstraction of actions in the form of primitive skills (e.g., options (Sutton et al., 1999)). However, significantly simplifying the problem for the higher level requires abstraction of both states *and* actions. This is particularly important with rich sensory observations, where standard options frameworks provide a greatly abstracted state space, but do not simplify the perception or estimation problem. The nature of the ideal state abstraction in HRL is closely tied to the action abstraction, as the most suitable abstraction of state should depend on the kinds of decisions that the higher-level policy needs to make, which in turn depends on the actions (skills) available to it. This presents a challenge in designing HRL methods, because it is difficult to devise a state abstraction that is both highly abstracted (and therefore removes many distractors) and still sufficient to make decisions for long-horizon tasks. This challenge differs markedly from representation learning problems in other domains, like computer vision and unsupervised learning (Schroff et al., 2015; van den Oord et al., 2018; Chen et al., 2020), since it is intimately tied to the *capabilities* exposed to the agent via its skills.

We therefore posit that a suitable representation for a higher-level policy in HRL should depend on the capabilities of the skills available to it. If this representation is sufficient to determine the abilities and outcomes of these skills, then the high-level policy can select the skill most likely to perform the desired task. This concept is closely tied to the notion of affordances, which has long been studied in cognitive science and psychology as an action-centric representation of state (Gibson,

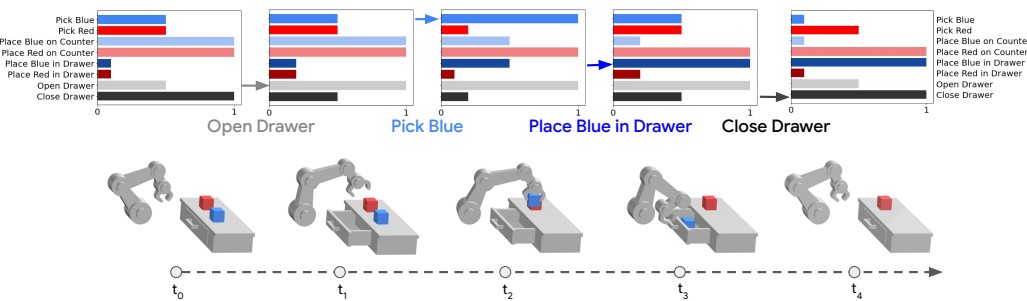

**(a)** A trajectory through the skill value function space for the task "Place blue block in drawer". VFS, visualized on top of corresponding scene, captures positional information about the contents of the scene, preconditions for interactions, and the effects of executing a feasible skill, making it suitable for high-level planning.

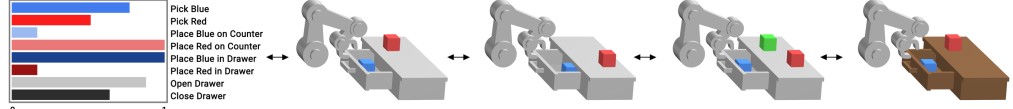

**(b)** VFS ignores task-irrelevant factors like arm pose, color of the desk, or distractors. All configurations shown above are *functionally equivalent* and hence, map to the same VFS representation.

**Figure 1:** Visualizing VFS embeddings in an example desk rearrangement task. VFS can capture the affordances of the low-level skills while ignoring exogenous distractors.

1977), and has inspired techniques in robotics and RL (Zech et al., 2017; Xu et al., 2021; Mandikal & Grauman, 2021). Given a set of skills that span the possible interactions in an environment, we propose that the value functions corresponding to these policies can provide a representation that suitably captures the capabilities of the skills in the current state and thus can be used to form a compact embedding space for high-level planning. We call this the Value Function Space (VFS). Note that we intend to use these skills as high-level actions without modifications—the important problems of skill discovery and test-time adaptation are beyond the scope of this paper.

Figure 1a illustrates the state abstraction constructed by VFS for the desk rearrangement example discussed above: VFS captures the affordances of the skills and represents the state of the environment, along with preconditions for the low-level skills, forming a functional representation to plan over. Since VFS constructs a skill-centric representation of states using the value functions of the low-level skills, it captures functional equivalence of states in terms of their affordances: in Figure 1b, states with varying object or arm positions, background textures, and distractor objects are functionally equivalent for planning, and map to the same VFS embedding. This simplifies the high-level planning problem, allowing the higher level policy to generalize to novel environments.

*Statement of Contributions.* The primary contribution of this work is VFS, a novel state representation derived from the value functions of low-level skills available to the agent. We show that VFS leverages the properties of value functions, notably their ability to model possible skills and completed skills, to form an effective representation for high-level planning, and is compatible with both model-free and model-based high-level policies. Empirical evaluations in maze-solving and robotic manipulation demonstrate that the skill-centric representation constructed by VFS outperforms representations learned using contrastive and information-theoretic objectives in long-horizon tasks. We also show that VFS can generalize to novel environments in a zero-shot manner.

## 2 RELATED WORK

Hierarchical RL has been studied extensively in the literature, commonly interpreted as a *temporal abstraction* of the original MDP. Early works have interpreted the hierarchy introduced in this setting as an abstraction of state and action spaces (Sutton et al., 1999; Dietterich, 2000; Thomas & Barto, 2012). The popular options framework Sutton et al. (1999); Precup (2000) provides a natural way of incorporating temporally extended actions into RL systems. An agent that possesses the transition model and reward model for such a SMDP (known as an *option model*) is capable of sample-based planning in discrete (Kocsis & Szepesvári, 2006) and continuous state spaces (Konidaris et al., 2014; Gopalan et al., 2017). However, doing so in environments with high-dimensional observations (such as images) is challenging. In this work, we explore the efficacy of learned skills operating on high-dimensional observations for long-horizon control in realistic environments.

To improve the quality of lower-level policies, recent work in HRL has studied various facets of the problem, including discovery of skills (Konidaris & Barto, 2009; Zhang et al., 2021b; Florensa et al., 2017; Warde-Farley et al., 2018), end-to-end training of both levels (Kulkarni et al., 2016; Tessler et al., 2017) and integrating goal-conditioned behaviors (Ghosh et al., 2019; Nachum et al., 2019). In this work, we assume that the low-level skills are given, and do not focus on discovering them. Instead, we focus on how skills can simplify the higher-level control problem by providing a representation that is suitable for either model-based or model-free control.

The problem of learning meaningful long-horizon behavior by combining a set of lower-level skills, or options, has been studied extensively in prior work. Popular approaches to composing skills involve combining them in the space of value functions (Todorov, 2009; Hunt et al., 2019; da Silva et al., 2009; Ziebart, 2010; Haarnoja et al., 2018), and more recently, using pseudo-rewards (Barreto et al., 2019). While these methods have been very effective at composing low-level skills, they do not address the question of learning a meaningful state abstraction using these skills. We take inspiration from these works and use the value function space to construct a skill-centric state abstraction, along with a high-level policy that composes these skills for temporally extended tasks.

Our method can hence be interpreted as a representation learning approach. Representation learning techniques have been employed extensively in model-free RL by augmenting auxiliary tasks based on reconstruction losses (Lange et al., 2012; Higgins et al., 2017; Yarats et al., 2019) and predicting the future conditioned on past observations (Schmidhuber, 1990; Jaderberg et al., 2017; van den Oord et al., 2018; Shelhamer et al., 2016). Contrastive learning has also been used in recent works to discover a meaningful latent space and extract reward signals for RL (Sermanet et al., 2017; Warde-Farley et al., 2018; Dwibedi et al., 2018; Laskin et al., 2020). Another popular approach to learning useful representation is to optimize for effective transfer of the lower-level skills and the learned state abstraction (Gupta et al., 2017; Konidaris & Barto; Goyal et al., 2019). Unlike these prior works, our aim is specifically to learn a representation that is grounded in the capabilities of the low-level skills, which gives us a skill-centric abstraction of high-dimensional observations. While we do not explicitly optimize for transfer like Gupta et al. (2017), our experiments show that VFS can generalize to novel environments while being robust to functionally-irrelevant distractors.

Alongside developments in model-free RL, prior work has also sought to learn predictive models of the environment for sampling and planning. This has been demonstrated by learning dynamics using future predictions (Watter et al., 2015; Oh et al., 2017; Ebert et al., 2017; Banijamali et al., 2018; Ha & Schmidhuber; Hafner et al., 2018; Ichter & Pavone, 2019; Hafner et al., 2020; Zhang et al., 2019), learning belief representations (Gregor et al., 2019; Lee et al., 2019) and representing state similarity using the bisimulation metric (Castro, 2020; Zhang et al., 2021a; Agarwal et al., 2021). The combination of learned model-free policies with structures like graphs (Savinov et al., 2018; Eysenbach et al., 2019) and trees (Ichter et al., 2021) to plan over extended horizons has also been demonstrated to improve generalization and exploration. While our primary objective is not to develop better model-based RL algorithms, we show that our proposed state abstraction can be utilized by model-based controllers to plan over temporally extended skills.

## 3 PRELIMINARIES

We assume that an agent has access to a finite set of temporally extended options $O$, or skills, which it can sequence to solve long-horizon tasks. These skills can be trained for a wide range of tasks by using manual reward specification (Huber & Grupen, 1998; Stulp & Schaal, 2011), using relabeling techniques (Andrychowicz et al., 2017), or via unsupervised skill discovery (Daniel et al., 2016; Fox et al., 2017; Warde-Farley et al., 2018; Sharma et al., 2020). Since many prior works have focused on skill discovery, we do not explicitly address how these skills are produced.

We assume that each skill has a maximum rollout length of $\tau$ time steps, after which it is terminated. We also assume that each skill $o_i \in O$ is accompanied by a critic, or value function, $V_{o_i}$ denoting the expected cumulative skill reward executing skill $o_i$ from current state $s_t$. This is generally available for policies trained with RL (Huber & Grupen, 1998; Andrychowicz et al., 2017; Kalashnikov et al., 2021) or learned via options discovery (Bacon et al., 2017; Klissarov et al., 2017; Tiwari & Thomas, 2019; Riemer et al., 2018b). While this may not be true for skills discovered by other methods, such as segmentation, graph partioning or bottleneck discovery (McGovern & Barto, 2001; Menache et al., 2002; Krishnamurthy et al., 2016; Şimşek & Barto, 2004; Şimşek et al., 2005; Riemer et al.,

2018a), we can use policy evaluation to automatically obtain a value function—this can be done via regression to empirical returns or temporal difference learning (Dann et al., 2014).

Our setting is closely related to the *options framework* of Sutton et al. (1999). Options are skills that consist of three components: a policy $\pi : \mathcal{S} \times \mathcal{A} \rightarrow [0, 1]$, a termination condition $\beta : \mathcal{S}^+ \rightarrow [0, 1]$, and an initiation set $\mathcal{I} \subseteq \mathcal{S}$, where $\mathcal{S}, \mathcal{A}$ are the low-level state and action spaces in a fully observable decision process. An option $\langle \mathcal{I}, \pi, \beta \rangle$ is available at state $s_t$ if and only if $s_t \in \mathcal{I}$. We do not assume we have an initiation set, and show that the value functions provide this information. We assume that the policies come with a termination condition, or alternatively, have a fixed horizon $\tau$. Next, we describe a framework to formulate a decision problem that uses these options for planning.

We assume that the low-level observation and action space of the agent can be described as a fully observable *semi-Markov decision process* (SMDP) $\mathcal{M}$ described by a tuple $(S, O, R, P, \tau, \gamma)$, where $S \subseteq \mathbb{R}^n$ is the $n$-dimensional continuous state space; $O$ is a finite set of temporally extended skills, or options, with a maximum horizon of $\tau$ time steps; $R(s'|s, o_i)$ is the task reward received after executing the skill $o_i \in O$ at state $s \in S$; $P(s'|s, o_i)$ is a PDF describing the probability of arriving in state $s' \in S$ after executing skill $o_i \in O$ at state $s \in S$; $\gamma \in (0, 1]$ is the discount factor. Given a finite set of options, our objective is to obtain a high-level decision-policy that selects *among* them.

Note that generally, many image-based problems are partially observed so that the entire history of observation-action pairs may be required to describe the state. Explicitly addressing partial observability is outside the scope of our work, so we follow the convention in prior work by assuming full observability and refer to images as states (Lillicrap et al., 2016; Nair et al., 2018).

## 4 SKILL VALUE FUNCTIONS AS STATE SPACES

The notion of value in RL is closely related to affordances, in that the value function predicts the capabilities of the skill being learned. The supervised learning problem of affordance prediction (Ugur et al., 2009) can in fact be cast as a special case of value prediction (Sutton & Barto, 2018; Graves et al., 2020). In this section, we construct a skill-centric representation of state derived from skill value functions that captures the affordances of the low-level skills, and empirically show that this representation is effective for high-level planning.

Given an SMDP $\mathcal{M}(S, O, R, P, \tau, \gamma)$ with a finite set of $k$ skills $o_i \in O$ trained with sparse outcome rewards and their corresponding value functions $V_{o_i}$, we construct an embedding space $Z$ by stacking these skill value functions. This gives us an abstract representation that maps a state $s_t$ to a $k$-dimensional representation $Z(s_t) := [V_{o_1}(s_t), V_{o_2}(s_t), ..., V_{o_k}(s_t)]$, which we call the Value Function Space, or VFS for short. This representation captures *functional information* about the exhaustive set of interactions that the agent can have with the environment by means of executing the skills, and is thus a suitable state abstraction for downstream tasks.

Figure 1(a) illustrates the proposed state abstraction for a conceptual desk rearrangement task with eight low-level skills. A raw observation, such as an image of the robotic arm and desk, is abstracted by a 8-dimensional tuple of skill value functions. This representation captures positional information about the scene (e.g., both blocks are on the counter and drawer is closed since the corresponding values are 1 at $t_0$), preconditions for interactions (e.g., both blocks can be lifted since the "Pick" values are high at $t_0$), and the effects of executing a feasible skill (e.g., the value corresponding to the "Open Drawer" skill increases on executing it at $t_1$), making it suitable for high-level planning.

We hypothesize that since VFS learns a skill-centric representation of the scene, it is robust to exogenous factors of variation, such as background distractors and appearance of task-irrelevant components of the scene. This also enables VFS to generalize to novel environments with the same set of low-level skill, which we demonstrate empirically. Figure 1(b) illustrates this for the desk rearrangement task. All configurations shown are *functionally equivalent* and can be interacted with identically. Intuitively, VFS would map these configurations to the same abstract state by ignoring factors like arm pose, color of the desk, or additional objects, which do not affect the low-level skills.

## 5 MODEL-FREE RL WITH VALUE FUNCTION SPACES

In this section, we instantiate a hierarchical model-free RL algorithm that uses VFS as the state abstraction and the skills as the low-level actions. We compare the long-horizon performance of VFS to alternate representations for HRL trained with constrastive and information theoretic objectives

for the task of maze-solving and find that VFS outperforms the next best baseline by up to 54% on the most challenging tasks. Lastly, we compare the zero-shot generalization performance of these representations and empirically demonstrate that the skill-centric representation constructed by our method can successfully generalize to novel environments with the same set of low-level skills.

## 5.1 An Algorithm for Hierarchical RL

We instantiate a hierarchical RL algorithm that learns a Q-function $Q(Z, o)$ at the high-level using VFS as the "state" $Z$ and the skills $o_i \in O$ as the temporally extended actions. Given such a Q-function, a greedy high-level policy can be obtained by $\pi_Q(Z) = \arg\max_{o_i \in O} Q(Z, o_i)$. We use DQN (Mnih et al., 2015), which uses mini-batches sampled from a replay buffer of transition tuples $(Z_t, o_t, r_t, Z_{t+1})$ to train the learned Q-function to satisfy the Bellman equation. This is done using gradient descent on the loss $\mathcal{L} = \mathbb{E} \left( Q(Z_t, o_t) - y_t \right)^2$, where $y_t = r_t + \gamma \max_{o_{t'} \in O} Q(Z_{t+1}, o_{t'})$.

In order to make the optimization problem more stable, the targets $y_t$ are computed using a separate target network which is update at a slower pace than the main network. Q-learning also suffers from an overestimation bias, due to the maximization step above, and harms learning. Hasselt et al. (2016) address this overestimation by decoupling the selection of action from its evaluation. We use this variant, called DDQN, for subsequent evaluations in this work. Note that we are using the skills as given, without updating or finetuning them.

While there have been several improvements to DDQN (Hessel et al., 2018) and alternative algorithms for model-free RL in discrete action spaces (Schaul et al., 2015; Bellemare et al., 2017; Christodoulou, 2019), and these improvements will certainly improve the overall performance of our algorithm, our goal is to evaluate the efficacy of VFS as a state representation and we study it in the context of a simple DDQN pipeline.

## 5.2 Evaluating Long-Horizon Performance in Maze-Solving

To evaluate the long-horizon performance of VFS against commonly used representation learning methods, we use the versatile MiniGrid environment (Chevalier-Boisvert et al., 2018) in a fully observable setting, where the agent receives a top-down view of the environment. We consider two tasks: (i) *MultiRoom*, where the agent is tasked with reaching the goal by solving a variable-sized maze spanning up to 10 rooms. The agent must cross each room and open the door to access the following rooms; (ii) *KeyCorridor*, where the agent is tasked with reaching a goal up to 7 rooms away and may face locked doors. The agent must find the key corresponding to a color-coded door and open it to access subsequent rooms. Both these tasks have a sparse reward that is only provided for successfully reaching the goal. This presents a challenging domain for long-horizon sequential reasoning, where tasks may require over 200 time steps to succeed, making a great testbed for evaluating the ability of the state abstractions to capture relevant information for sequencing multiple skills. The agents have access to the following temporally extended skills—GoToObject, PickupObject, DropObject and UnlockDoor—where the first three skills are text-conditioned and Object may refer to a door, key, box or circle of any color. Since Mini-Grid is easily reconfigurable, we also generate a set of holdout *MultiRoom* mazes with different grid layouts to evaluate the zero-shot generalization performance of the policies trained with these representations. Note that the grid layouts, as well as object positions, for these tasks are randomly generated for every experiment and are not static. Example grid layouts are shown in Figure 2. We provide further implementation details in Appendix A.1.

**Baselines:** We compare VFS extensively against a variety of competitive baselines for representation learning in RL using contrastive and information-theoretic objectives. We consider representations learned both offline and online (in loop with RL); note that VFS is constructed entirely from the values of the available skills and is not learned. Further, all baselines have access to these skills.

1. *Raw Observations:* We train a high-level policy operating on raw input observations.
2. *Autoencoder (AE):* We use an autoencoder to extract a compact latent space using a reconstruction loss on an offline dataset of trajectories, similar to Lange et al. (2012).
3. *Contrastive Predicting Coding (CPC):* We learn a representation by optimizing the InfoNCE loss over an offline dataset of trajectories (van den Oord et al., 2018).

| Representation | MultiRoom | | | | KeyCorridor | |
|---|---|---|---|---|---|---|
| | **2** | **4** | **6** | **10** | **3** | **7** |
| Raw Observations | 0.64 | 0.46 | 0.42 | 0.29 | 0.47 | 0.32 |
| AE (Lange et al., 2012) | 0.70 | 0.64 | 0.51 | 0.34 | 0.59 | 0.33 |
| CPC (van den Oord et al., 2018) | 0.77 | 0.69 | 0.55 | 0.37 | 0.63 | 0.35 |
| VAE[†] (Yarats et al., 2019) | 0.79 | 0.74 | 0.58 | 0.49 | 0.79 | 0.50 |
| CURL[†] (Laskin et al., 2020) | 0.82 | 0.76 | 0.63 | 0.43 | **0.83** | 0.54 |
| VFS (Ours) | **0.98** | **0.92** | **0.83** | **0.77** | 0.82 | **0.68** |

**Table 1:** Success rates of different representations for model-free RL across varying levels of difficulty and time horizons (second row denotes complexity in terms of number of rooms). VFS explicitly captures the capabilities of the low-level skills, and outperforms all baselines. Online methods (denoted by [†]) learned jointly with the RL objective outperform their offline counterparts (AE and CPC), but their performance degrades for longer-horizon tasks. Refer to Appendix B for additional performance metrics and ablations.

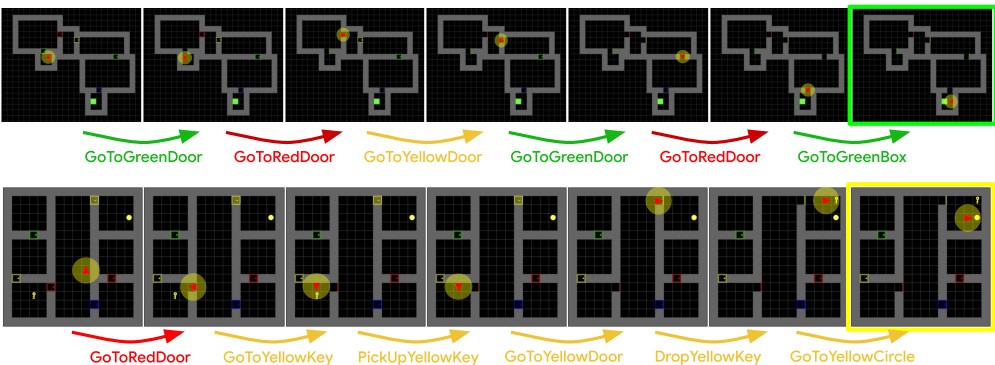

**Figure 2:** Successful rollouts of HRL using VFS to solve long-horizon tasks in *MultiRoom-6* (**top**) and *KeyCorridor-7* (**bottom**) by sequentially executing multiple low-level skills (labeled under the arrows).

4. *Online Variational Autoencoder (VAE):* We learn a VAE representation jointly (online) with the high-level policy operating on this representation (Yarats et al., 2019).

5. *Contrastive Unsupervised Representations for RL (CURL):* We learn a representation by optimizing a contrastive loss jointly (online) with the RL objective (Laskin et al., 2020).

**Evaluation:** We run experiments for the two tasks with varying number of rooms, to study the performance of the algorithms with increasing time horizon, and report the success rates in Table 1. HRL from raw observations demonstrates a success rate of 64% in the two-room environment, which can be attributed to the powerful set of skills available to the high-level policy, but this quickly drops to 29% in the largest environment. The offline baselines (AE and CPC) construct a compact state abstraction, which makes the learning problem easier for DDQN, and show significant improvements in smaller environments (MultiRoom-2,4 and KeyCorridor-3). However, they are unable to improve the performance in larger environments. We hypothesize that this is due to the inability of the representations to capture information necessary for the high-level policy, since they are learned independent of the controller. The performance of representations learned online (VAE and CURL), which are implemented analogous to their offline counterparts (AE and CPC, respectively) support this hypothesis and improves the performance across all tasks by learning representations jointly with the controller, and scoring up to 54% in the most challenging environment. We hypothesize that the limited performance of these methods is due to the lack of direct influence of the downstream task on the representation. Despite being constructed offline, VFS explicitly captures the capabilities of the low-level skills and provides an action-centric representation for the high-level policy, greatly simplifying the control problem. This is reflected in the performance of VFS across all tasks—it outperforms all baselines, scoring 98% on the simplest task and up to 68% on the most challenging task, beating the next best method by over 25%. Figure 2 shows sample rollouts of the high-level policy using VFS as the state representation in the two tasks discussed. It is important to note that while VFS outperforms the baselines in these tasks, our method is largely orthogonal to these representation learning algorithms and can be combined constructively to improve task performance—we show this, along with further ablations in Appendix B.

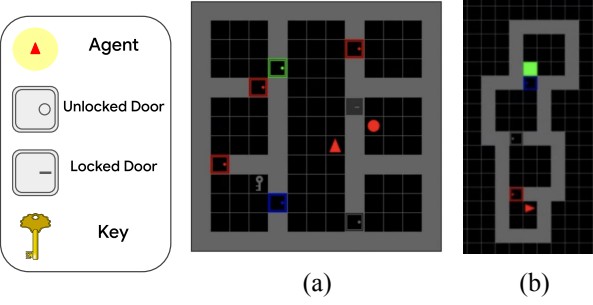

| Representation | MR4 | MR10 |
|---|---|---|
| Raw | 0.15 | 0.03 |
| AE | 0.41 | 0.23 |
| CPC | 0.47 | 0.20 |
| VAE[†] | 0.25 | 0.03 |
| CURL[†] | 0.27 | 0.07 |
| VFS (Ours) | **0.87** | **0.67** |
| HRL-Target | 0.92 | 0.77 |

**Figure 3:** We study the zero-shot generalization by using policies trained in `KeyCorridor` **(a)** and deploying in `MultiRoom` **(b)**.

**Table 2:** Success rates for zero-shot generalization. VFS can generalize zero-shot to novel environments.

### 5.3 ZERO-SHOT GENERALIZATION

We evaluate the generalization abilities of these representations by training them for the *KeyCorridor* task (as above) and evaluating on *MultiRoom*. Since the skills available in *MultiRoom* are a subset of those in *KeyCorridor*, we ignore any invalid actions executed by the agent. Note that the high-level agent has not been trained in *MultiRoom* and the policies are not updated in the target environments.

Table 2 shows the success rates of the representations in the *MultiRoom* tasks with 4 (*MR4*) and 10 rooms (*MR10*). Unsurprisingly, HRL with raw observations fails to generalize, because the maze layout differs significantly (e.g. see Figure 3). AE and CPC learn a compact representation from high-dimensional observations and allow the high-level policy to generalize to simpler tasks like *MR4* and achieve up to 47% success rate. Interestingly, their online counterparts (VAE and CURL) also fail to generalize and perform poorly, likely because representations learned jointly with the RL policy can overfit to the source environment. VFS learns a skill-centric representation that can generalize zero-shot to tasks that use the same skills, and thus outperforms the next best baseline by up to 180%, closely matching the performance of an HRL policy trained from scratch in the target environment with online interaction (*HRL-Target*).

## 6 MODEL-BASED PLANNING WITH VALUE FUNCTION SPACES

In this section, we introduce a simple model-based RL algorithm that uses VFS as the "state" for planning, which we term VFS-MB. We study the performance of VFS in the context of a robotic manipulation task, and compare it to alternate representations for model-based RL. We find that the performance of VFS as an abstract representation for raw image observations outperforms all baselines and closely matches that of a pipeline with access to oracular state of the simulator, showing the efficacy of our method in long-horizon planning.

### 6.1 A SIMPLE ALGORITHM FOR MODEL-BASED RL

We use a simple model-based planner that learns a *one-step* predictive model, using VFS as the "state." Specifically, this model learns the transition dynamics $Z_{t+1} = \hat{f}(Z_t, o_t) \, \forall \, o_t \in O$ via supervised learning using a dataset of prior interactions in the environment. Note that the predictive model (and the subscript of $Z$) is over high-level policy steps, which may be up to $\tau$ steps of the low-level skills. This dataset can be collected simply by executing the available skills in the environment for $\tau$ steps, where $\tau$ is the maximum horizon of the SMDP $\mathcal{M}$, or until termination.

In order to use the learned model $\hat{f}(Z_t, o_t)$, a goal latent state $Z_g$, and a scoring function $\epsilon$ (e.g. mean squared error) for the high-level task, we need to solve the following optimization problem for the optimal sequence of skills $(o_t, \ldots, o_{t+H-1})$ to reach the goal:

$$(o_t, \ldots, o_{t+H-1}) = \operatorname*{arg\,min}_{o_t,\ldots,o_{t+H-1}} \epsilon(\hat{Z}_{t+H}, Z_g) : \hat{Z}_t = Z_t, \ \hat{Z}_{t'+1} = \hat{f}(\hat{Z}_{t'}, o_{t'}) \qquad (1)$$

We use a sampling-based method to find solutions to the above equation. We use random shooting (Rao, 2009) to randomly generate $K$ candidate option sequences, predict the corresponding $Z$ sequences using the learned model $\hat{f}$, compute the rewards for all sequences, and pick the candidate action sequence leading to a latent state closest to the goal, according to Equation 1. We execute the

policy using model-predictive control: the policy executes only the first skill $o_t$, receives the updated $Z_{t+1}$ from the environment, and recalculates the optimal action sequence iteratively. Figure 4 shows an overview of the algorithm. We provide further implementation details in Appendix A.2

Note that our goal is to study the behavior of VFS as a state representation in existing RL pipelines, rather than developing a better model-based algorithm. Using sophisticated methods that adjust the sampling distribution, as in the cross-entropy method (Finn & Levine, 2017; Hafner et al., 2018; Chua et al., 2018) or path integral optimal control (Williams et al., 2015; Lowrey et al., 2019), as well as more sophisticated models and planning or control methods would likely improve overall performance further, but is outside the scope of this work.

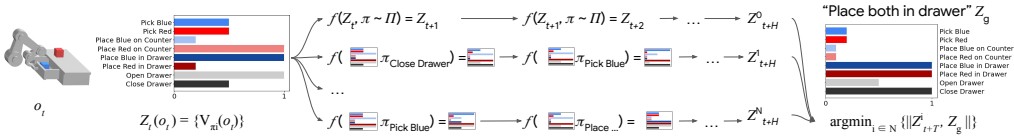

**Figure 4:** Overview of model-based planning with VFS. We learn a one-step predictive model for the embedding $Z_t$ and use random shooting with a scoring function $\epsilon$ to find the best action to the encoded goal $Z_g$.

## 6.2 APPLICATION: ROBOTIC MANIPULATION

We evaluate the performance of VFS-MB in a complex image-based task using a simulated manipulation environment with an 8-DoF robotic arm, similar to the setup used by Jang et al. (2021). The robot only has access to high-dimensional egocentric visual observations. We consider the task of semantic rearrangement, which requires rearranging objects into semantically meaningful positions—e.g., grouped into (multiple) clusters, or moved to a corner (Figure 5). Efficiently solving this task requires the robot to plan over a series of manipulation maneuvers, interacting with up to 10 distinct objects. This requires long-horizon planning, and presents a major challenge for RL methods. Each of the methods we compare have access to a library of MoveANearB skills, trained as a multi-task policy with MT-Opt (Kalashnikov et al., 2021). There are 10 objects, and 9 possible destinations near which they can be moved, resulting in a total of 90 skills, which then comprise the action space for planning. The skills control the robot at a frequency of 0.5 Hz, taking an average of 14 time steps to execute with a success rate of 94%. We provide further details about these skills in Appendix A.2.

We consider two versions of the task, which arrange either 5 or 10 objects to semantic positions. The *O5* environment uses a random subset of the 10 objects in every experiment; lesser objects allow a smaller planning horizon, making planning problem simpler than in the *O10* environment, which has all 10 objects. We randomize the object positions on the table and command the algorithms to reach the same goal with a planning horizon $H = 7$ steps for the smaller environment and $H = 15$ steps for the larger environment, reporting the task success rate averaged over 20 experimental runs.

**Baselines:** To evaluate the efficacy of the proposed representation for high-level model-based planning, we compare it against four alternative representations used in conjunction with the algorithm described in Section 6.1. All methods have access to the skills and use them as the low-level action space. *Raw Image* learns a policy on raw visual observations from the robot's onboard camera. *VAE* uses a variational autoencoder to project the observations to a $100-$dimensional learned embedding space, similar to Corneil et al. (2018). We train this VAE offline from trajectories collected by rolling out the models. *CPC* uses contrastive predictive coding (van den Oord et al., 2018) to learn a similar representation from offline trajectories. We also compare against an *Oracle State* baseline that has access to privileged information—the simulator state—and learns the model on this. This gives us an upper bound for the performance of the baselines.

**Evaluation:** Table 3 shows the success rates on the two versions of the task for each prior method. The model trained using image observations fares poorly and fails at all but the simplest starting configurations. It succeeds in 2/20 experiments in *O10*, largely due to poor model predictions. The VAE and CPC representations, which learn a more compact embedding space to plan over, succeed in up to 65% of the tasks in the simpler *O5* environment. However, their performance falls sharply in

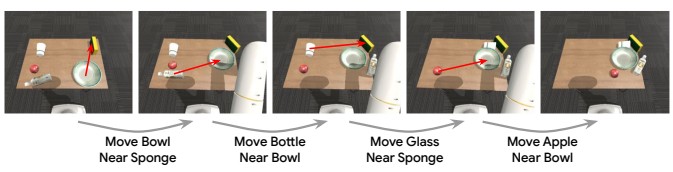

Move Bowl    Move Bottle    Move Glass    Move Apple
Near Sponge   Near Bowl    Near Sponge   Near Bowl

**Figure 5:** Example rollout of model-based RL with VFS as the state representation for robotic manipulation. The robot plans over multiple low-level skills to achieve the semantic goal "move all to top-right corner". Red arrows specify the next skill planned by the model, and is overlaid for visualization purposes only.

| Representation | O5 | O10 |
|---|---|---|
| Raw Image | 0.25 | 0.1 |
| VAE | 0.6 | 0.3 |
| CPC | 0.65 | 0.4 |
| VFS (Ours) | **1** | **0.8** |
| Oracle State | 1 | 0.85 |

**Table 3:** Success rates for robotic manipulation. VFS outperforms all baselines and closely matches oracular performance.

the *O10* environment, where the longer horizons read to greater error accumulation. VFS constructs an effective representation that captures the state of the scene as well as the affordances of the skills, and succeeds in all tasks in the *O5* environment, matching the oracle performance. It also succeeds in 80% of the tasks in *O10*, closely matching the oracle's performance of 85%.

To understand the factors captured by VFS, we sample encoded observations from a large number of independent trajectories and visualize their 2D t-SNE embeddings (van der Maaten & Hinton, 2008). Figure 6 shows that VFS can successfully capture information about objects in the scene and affordances (e.g. which object is in the robot's arm and can be manipulated), while ignoring distractors like the poses of the objects on the table and the arm.

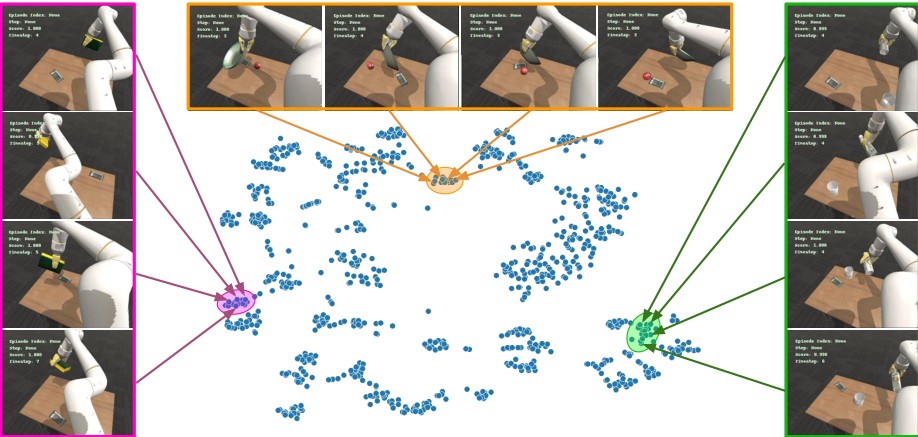

**Figure 6:** A t-SNE diagram of observations encoded by VFS, showing functionally equivalent observations mapped to the same representation. VFS discovers clusters with **(top)** the arm grasping the bowl with apple and chocolate on the table, **(right)** bottle in arm with glass and chocolate on table; **(left)** sponge in arm with chocolate on the table. Note that these observations occur across independent trajectories and are unlabeled.

## 7 DISCUSSION

We proposed Value Function Spaces as state abstractions: a novel skill-centric representation that captures the *affordances* of the low-level skills. States are encoded into representations that are *invariant* to exogenous factors that do not affect values of these skills. We show that this representation is compatible with both model-free and model-based policies for hierarchical control, and demonstrate significantly improved performance both in terms of successfully performing long-horizon tasks and in terms of zero-shot generalization to novel environments, which leverages the invariances that are baked into our representation.

The focus of our work is entirely on *utilizing* a pre-specified set of skills, and we do not address how such skills are learned. Improving the low-level skills jointly with the high-level policy could lead to even better performance on particularly complex tasks, and would be an exciting direction for future work. More broadly, since our method connects skills directly to state representations, it could be used to turn unsupervised skill discovery methods directly into unsupervised representation learning methods, which could be an exciting path toward more general approaches that retain the invariances and generalization benefits of our method.

## ACKNOWLEDGMENTS

This research was partially supported by ARL DCIST CRA W911NF-17-2-0181. The authors would like to thank Matthew Benice for help with setting up and debugging the robotic manipulation environment, and Ryan Julian for training the skills used in our experiments. The authors would also like to thank Sumeet Singh, Katie Kang and Kyle Hsu for feedback on an earlier draft of this paper.

## REPRODUCIBILITY STATEMENT

The primary contribution of our work, VFS, is a skill-centric representation designed to work with existing model-based and model-free pipelines. The implementation simply involves concatenating the value functions corresponding to the available skills into an embedding vector that can be used for HRL or planning. We provide all necessary information for setting up VFS to work with a reader's RL algorithm of choice in Sections 5.1, 6.1 and Appendix A. Our model-free experiments are conducted on an open-source gym environment and we provide configuration details for setting up our quantitative experiments in Appendix A.1. We hope that this encourages the community to utilize and build upon the ideas presented in the paper. We plan to release more information about the proprietary environments and tasks used in a public release[1] of this article at a later date.

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
