# OpenReview forum: "Value Function Spaces: Skill-Centric State Abstractions for Long-Horizon Reasoning"
_ICLR.cc/2022/Conference — ICLR 2022 Poster_

### Official Review · Reviewer_8FMV · 2021-11-01

**Correctness:** 3
**Technical Novelty And Significance:** 3
**Empirical Novelty And Significance:** 3
**Recommendation:** 6
**Confidence:** 3

**Main Review:**

# Strengths:

(1) The idea of using the value estimates of different pre-existing policies as state representation is, to my best knowledge, novel.

(2) Since the value estimates of pre-trained policies are used directly as states of the higher-level policy, the method can make use of the large set of pre-existing policies available in the wild.

(3) The paper is generally well written, and the narrative is easy to follow.

# Weaknesses:

(1) I have the feeling that the comparison with the baselines is not entirely fair. The baselines have to learn a good state embedding and a good higher-level policy simultaneously, while the proposed method only needs to learn the higher-level policy. However, this completely ignores the fact that the proposed method had access to a much larger corpus of transition since it uses the pre-trained value functions. In contrast, the baselines do not have access to the value functions of the skills.

(2) I would like to see the method evaluated in much more environments. From the existing experiments, it is hard to judge how the method performs in different settings.

# Needs clarification and general comments

(1) How does this method deal with different scalings of the value estimates? Especially in the dense reward case, value estimates for different skills can differ significantly, i.e. because of reward shaping.

(2) Paragraphs 2 and 3 in section 4 repeat more or less what is written in paragraph 4 of the introduction.

(3) To me, it does not feel right to artificially restrict the embedding spaces of the baselines to the maximum size of the VFS embedding space.
What is the justification for that? What are the best-case results for the baselines without that restriction?

(4) In the case of the offline baselines: How many transitions were seen during training of the embedding space compared to the number of transitions used for learning the different skills/policies and their respective value functions?
Similarly, in the case of the online baselines: How many transitions were used in total (pre-training plus training of the higher-level policy) to train the proposed method compared to train the baselines (which do not have access to the value function estimates of the skills)?

# Final decision due to rebuttal:

The authors addressed many of my concerns and provided new experimental results that led to interesting new insights. Hence, I am happy to raise my score from 5 to 6.

**Summary Of The Paper:**

The paper proposes a state abstraction in hierarchical reinforcement learning (HRL), called Value Function Spaces (VFS), that is constructed from the value functions of task-conditioned lower-level policies (or skills). The idea is that the value functions capture affordances of the lower-level skills while ignoring task-irrelevant information.
The lower-level policies and the corresponding value functions are given a prior, and the state is constructed by concatenating the value estimate of each lower-level policy.
The state abstraction is evaluated in a model-free Q-learning and model-based MPC type scenario.

**Summary Of The Review:**

The idea of using the value estimates of pre-trained policies is interesting and the empirical results look promising. However, as I pointed out in the weaknesses section, I do not think that the comparison with the baselines is entirely fair. Additionally, I would like to see the method evaluated in more environments. Because of that, I tend towards a weak reject, but I am willing to change my score if the authors can address these issues.

---

> ### Author Response · Authors · 2021-11-17
> **Response to Reviewer 8FMV: new experiments and discussion**
>
> Thank you for your insightful comments and feedback. We address the concerns raised by the reviewer below:
>
> > The comparison with the baselines is not entirely fair.
>
> While we think the reviewer is in agreement on this, but we still want to clarify that all baselines in our experiments have access to the same low-level set of skills. We want to emphasize that the ability to use pre-existing value functions in a clever way to derive a skill-cetric abstraction for states -- being able to do so without having to define an explicit representation learning objective or training pipeline -- is a unique strength of the proposed method. Our experiments show that other methods that don’t use these values are unable to learn a good representation using one of the mutual information or contrastive metrics.
>
> Furthermore, these baselines are not in opposition to VFS and can be constructively combined through learning a state space of (VFS + Image) passes through a VAE/CPC etc. We are running experiments to share an empirical analysis of this, please let us know if there are other baselines or competing methods that we should discuss or compare against.
>
> We believe the current experimental domains show the versatility of VFS to challenging long-horizon tasks in robotics and reinforcement learning, both in existing model-based and model-free pipelines, but we would request the reviewer share any suggestions on suitable additional environments, particularly for future work where we hope to focus on instruction-following, partial observability and discovering/improving low-level skills jointly with the high-level.
>
> > How does it deal with scalings in value estimates?
>
> For values obtained from different sources, we scale them to [0, 1] so they can be compatible with each other. We have not explored situations where the values are vastly different due to reward shaping, but we expect VFS to handle them fine as long as the training and test environments have similar skill-value pairs because the high-level policy or model is trained with data for the specific policies.
>
> > Experiments on increasing the embedding space of baselines.
>
> The choice of keeping the embedding space consistent was arbitrary and seems fair, so that the high-level policy operates on the same “amount of information”. For instance, allowing the VAE to have an embedding comparable to the input image would allow it to perform very well in simple tasks (no representation learning required), but fail at generalizing in presence of distractors because the representation may overfit to the input observations.
> We ran the model-free experiments with 2x and 5x embedding size and find that the performance of some baselines improves marginally:
>
> | Representation | MultiRoom-10 | KeyCorridor-7 |
> |:--------------:|:------------:|:-------------:|
> |   VFS (Ours)   |     0.77     |      0.68     |
> |    CURL (1x)   |     0.43     |      0.54     |
> |    CURL (2x)   |     0.47     |      0.56     |
> |    CURL (5x)   |     0.48     |      0.56     |
> |    VAE (1x)    |     0.58     |      0.50     |
> |    VAE (2x)    |     0.61     |      0.51     |
> |    VAE (5x)    |     0.62     |      0.53     |
> |     AE (1x)    |     0.34     |      0.33     |
> |     AE (2x)    |     0.33     |      0.37     |
> |     AE (5x)    |     0.37     |      0.40     |
>
> We will also run experiments increasing beyond this and include the results in the Appendix.
>
> > Number of transitions for offline and online baselines.
>
> For the offline baselines, we use on the order of 3M transitions for learning the embedding space which are similar to those used for learning the skills and values for VFS (2-3M, varies slightly across the skills). This is followed by about 10K high-level transitions for learning the HRL policy, giving a total of about 1M additional transitions (max 100 low-level transitions per high-level). For the online baselines, we allow significantly more samples since they are expected to learn the embedding as well as high-level policies. We use a total of 10M (+- 10%) transitions in this case.
>
> We thank the reviewer for pointing this out and we are updating the manuscript to include these details in the Appendix, as they are key to justifying the fairness of comparison against baselines.
>
> > Repetitive text in Section 4.
>
> We thank the reviewer for pointing this out, we are updating the writing to improve this.

---

> > ### Comment · Reviewer_8FMV · 2021-11-22
> > **Comments on authors repsond.**
> >
> > I thank the authors for the detailed response.
> >
> > > Furthermore, these baselines are not in opposition to VFS and can be constructively combined through learning a state space of (VFS + Image) passes through a VAE/CPC etc. We are running experiments to share an empirical analysis of this, please let us know if there are other baselines or competing methods that we should discuss or compare against.
> >
> > Yes, these would indeed be interesting baselines
> >
> > > We believe the current experimental domains show the versatility of VFS to challenging long-horizon tasks in robotics and reinforcement learning, both in existing model-based and model-free pipelines, but we would request the reviewer share any suggestions on suitable additional environments, particularly for future work where we hope to focus on instruction-following, partial observability and discovering/improving low-level skills jointly with the high-level.
> >
> > My comment was more going into the direction of what was also mentioned by reviewer aM2S: I would like to see a more thorough experimental evaluation of the claims made in the paper regarding the robustness of the method and the invariance to distractors. So far, the authors tested their method in two (similar) discrete and one continuous environment.
> >
> > >  We ran the model-free experiments with 2x and 5x embedding size
> >
> > Great, thanks for providing these experiments.

---

> > > ### Author Response · Authors · 2021-11-23
> > > **Continuing conversation: new experiments**
> > >
> > > We thank the reviewer for their timely response and sharing more comments and feedback.
> > >
> > > > New experiments
> > >
> > > We have conducted additional experiments that demonstrate that VFS can be used constructively with other representation learning methods, as per the previous round of our discussions. While the relative ordering of the results was not entirely surprising, we learned some new, interesting behavior about our method.
> > >
> > > When compared to the original baselines, adding VFS to the learned embedding significantly improves its performance (upto 100% boost). However, when comparing the added baselines to vanilla VFS, we observe minor improvements (upto 9%) which are statistically insignificant in some cases. This suggests that most of the heavylifting is done by the VFS state, as compared to the VAE or CPC embeddings. The results preserve relative ordering, which suggests that any advancement in the state of representation learning may be able to improve the overall performance of VFS when combined.
> > >
> > > | Representation | MultiRoom-4 | MultiRoom-10 | KeyCorridor-3 | KeyCorridor-7 |
> > > |:--------------:|-------------|:------------:|---------------|:-------------:|
> > > |       Raw      |     0.42    |     0.23     |      0.41     |      0.23     |
> > > |       AE       |     0.57    |     0.29     |      0.52     |      0.24     |
> > > |       CPC      |     0.62    |     0.34     |      0.55     |      0.27     |
> > > |       VAE      |     0.62    |     0.42     |      0.73     |      0.42     |
> > > |      CURL      |     0.66    |     0.36     |      0.72     |      0.45     |
> > > |       VFS      |     0.84    |     0.68     |      0.72     |      0.59     |
> > > |    VFS + Raw   |     0.83    |     0.69     |      0.71     |      0.58     |
> > > |    VFS + AE    |     0.83    |     0.68     |      0.73     |      0.57     |
> > > |    VFS + VAE   |     0.88    |     0.73     |      0.74     |      0.65     |
> > > |    VFS + CPC   |     0.90    |     0.71     |      0.77     |      0.63     |
> > >
> > > > Claims of robustness of the method and the invariance to distractors
> > >
> > > We make 2 key claims about our algorithm in the paper: (i) VFS is a versatile and effective way to learn high-level policies and state abstractions for solving long-horizon tasks, and (ii) state abstractions learned with VFS are generalizable to novel environments. We also remark that (and we have toned down these _claims_ in response to reviewer aM2S’ concerns) (iii) the representations learned by VFS are robust to distractors.
> > >
> > > Towards these claims, we present model-based and model-free pipelines and evaluate our algorithm against competitive baselines in Sections 5.2 and 6.2 (towards (i)), we present zero-shot generalization experiments in Section 5.3 (towards (ii)) that show that most other baselines perform very poorly under the slightest of changes in their observation space while VFS is able to maintain its performance and generalize in a zero-shot manner, and show qualitatively, that VFS is able to maintain an invariance to distractors like arm pose and background objects in the robotic manipulation task in Figure 6 (towards (iii)). We understand that the last point is only qualitative and have updated the manuscript in several places to clarify that this is a qualitative observation, rather than a theoretically-motivated claim, which is an exciting emergent behavior in its own right.
> > >
> > > We would be happy to add additional experiments for any claims that you feel have not been adequately validated for the final version (unfortunately, this late in the rebuttal process, we cannot add additional experiments, since revisions cannot be made after today). Are there any specific experimentational evaluations you would request, or particular claims that you feel have not been well validated?

---

> > > > ### Comment · Reviewer_8FMV · 2021-11-29
> > > > **Final decision**
> > > >
> > > > The authors addressed many of my concerns and provided new experimental results that led to interesting new insights. Hence, I am happy to raise my score from 5 to 6.

---

> ### Author Response · Authors · 2021-11-27
> **Checking back**
>
> Dear Reviewer,
>
> We hope that you've had a chance to read our response. We would really appreciate a reply as to whether our response and clarifications have addressed the issues raised in your review, or whether there is anything else we can address.

---

### Official Review · Reviewer_78KW · 2021-11-01

**Correctness:** 4
**Technical Novelty And Significance:** 2
**Empirical Novelty And Significance:** 3
**Recommendation:** 6
**Confidence:** 3

**Main Review:**

This paper presents an interesting approach by which one can get more out of trained skills; their primary contribution (the novel VFS representation) is easy to construct for skill-driven agents and could therefore serve as a relatively general representation to allow for performance improvement across a range of applications in which skills are already being used to make progress. Overall, though the idea is somewhat simple, it seems to me to be novel and has wide-ranging applications, including a number of applications demonstrated here.

My biggest concern is the lack of performance metrics beyond simply "success rate". Their omission is somewhat surprising and the authors should either include these additional performance metrics or discuss why they are unnecessary. The zero-shot generalization results are indeed promising, but it is hard to really evaluate how well the VFS agent performed compared to the HRL-Target agent (which is trained directly on this problem).
"Success-weighted path length" (SPL), a metric used by the Vision-and-Language community, would be somewhat appropriate, though presenting the "average performance/cost of successful trials" would perhaps be more useful. While I am not sure that this addition is *essential* for publication, these results are rather important for understanding the value of the approach and so the authors should make an effort to include them. The authors should either add this metric or adequately justify its omission.

Relatedly, it would be helpful to understand the nature of the failure cases: What does a failure look like? Some understanding of when and why the system fails is essential to understanding where future work is needed. For instance, in Sec 5.2, the agent fails in 32% of the more challenging environments: when it fails, is this a limitation of the low-level skills, of the ability of the agent to correctly predict the transition model, or both? Even if the authors do not readily have the answers to these questions, a few sentences commenting on the nature of experiments that would be necessary to answer them would be a welcome addition to the paper and help to improve the understanding of both readers and those that would build upon this approach. Similarly, what are the limitations of this approach? When would we expect that it would not be effective or fail? The Discussion section touches upon some of the potential directions for future improvement, but additional comments on the limitations of this version would help understanding. Additional experiments are not necessary here (nor are specific examples); a high-level discussion of these questions will suffice.

Questions and smaller suggestions:
- [Sec 6.1] Where does the "goal latent sate" come from? My intuition is that there would a non-latent goal state provided to the planner, which would be subsequently "encoded" into its VFS form. Is that correct? Either way, this should be clarified in the paper (along with, perhaps, a more comprehensive overview of the planning procedure).
- [Figure 1(b)] Though it is mostly clear from context, using an "equal" sign is not quite correct here, since the states themselves are in fact different. It is certainly possible to understand what the authors are going for, but changing the figure to reflect that applying the VFS encoding function makes all these states equal to one another would be a more appropriate way of communicating this knowledge, especially important since this figure is supposed to outline the core of the authors approach.
- [Figure 6] What are the skills available to the agent in this environment? Adding a few examples would help the reader understand the clusters better, though the idea that "the pose of objects is distractor information" is already clear from the figure.

**Summary Of The Paper:**

This paper presents Value Function Spaces (VFS), an abstract "skill-centric" representation for reinforcement learning and long-horizon planning. The core idea of VFSs is to leverage the learned value functions that are often trained alongside skills during skill learning to abstract knowledge about the world. Each element of this low-dimensional representation corresponds to the value estimated from one of the learned value function, so that the representation (by construction) tends to ignore distractor information unnecessary for accomplishing the skills. Using this representation on a set of pre-trained skills, the authors demonstrate improved performance on long-horizon planning tasks in grid-based maze and "locked door" environments, showing improved success probability on a number of such tasks compared to competitive baselines. They also show the ability of their abstraction to facilitate model-based goal-directed planning by learning a state-transition model (in VFS-space) that allows one to predict the outcome of executing a skill.

**Summary Of The Review:**

This paper presents an interesting approach by which one can get more out of trained skills; their primary contribution (the novel VFS representation) is easy to construct for skill-driven agents and could therefore serve as a relatively general representation to allow for performance improvement across a range of applications in which skills are already being used to make progress. One of my primary concerns is the lack of inclusion of cost-based performance metrics for their experiments, which I feel are somewhat important for understanding the performance of the approach. Even without them, I think this work represents a (mostly) well-executed, interesting idea, and so I am generally in favor of publication.

---

> ### Author Response · Authors · 2021-11-17
> **Response to Reviewer 78KW: new experiments, SPL performance metric**
>
> Thank you for the encouraging comments and feedback. We appreciate that you found this work “interesting” and “relatively general”. We address your concerns below:
>
> > Additional performance metrics.
>
> That is a great suggestion! We have run our experiments again and evaluated the SPL for the minigrid environment below. We’re not sure of how to design an equivalent metric for the robotic manipulation domain, please let us know if you have any pointers.
>
>
> | Representation | MultiRoom-4 | MultiRoom-10 | KeyCorridor-3 | KeyCorridor-7 |
> |:--------------:|-------------|:------------:|---------------|:-------------:|
> |       Raw      |     0.42    |     0.23     |      0.41     |      0.23     |
> |       AE       |     0.57    |     0.29     |      0.52     |      0.24     |
> |       CPC      |     0.62    |     0.34     |      0.55     |      0.27     |
> |       VAE      |     0.62    |     0.42     |      0.73     |      0.42     |
> |      CURL      |     0.66    |     0.36     |      0.72     |      0.45     |
> |   VFS (Ours)   |     0.84    |     0.68     |      0.72     |      0.59     |   Oracle HL   |     0.88    |     0.74     |      0.76     |      0.66     |
>
> We also added a new oracle baseline that does perfect planning at the high-level. We observe that the relative performance of the methods is largely unchanged, and the imperfect nature of skills causes a 10-20% reduction in SPL values across the board. When comparing performance to the oracular baseline, we notice that VFS achieves a performance close to the oracle, suggesting that the common failure cases are related to the imperfect skills and not due to planning.
>
>
>
> > What is the nature of failure cases?
>
> The low-level skills available to VFS are learned and not necessarily perfect. We find that a major portion of the failure cases are due to the lower-level skills not executing perfectly, causing the high-level planner to get stuck in a situation where it repeatedly executes the skills awaiting success (upto 80% of the failure cases). A small number of failures can also be attributed to bad planning, likely due to model errors. We try to capture this quantitatively by adding a new oracle baseline with “perfect planning”: We use a human expert as HL planner to obtain this baseline. This denotes a “perfect” high-level policy. The failure cases with this baseline denote the cases where the imperfect low-level skills led to failure.
>
> Maze-solving, model-free HRL
>
> | Representation | MultiRoom-10 | KeyCorridor-7 |
> |:--------------:|:------------:|:-------------:|
> |       AE       |     0.34     |      0.33     |
> |       CPC      |     0.37     |      0.35     |
> |       VAE      |     0.58     |      0.50     |
> |      CURL      |     0.43     |      0.54     |
> |   VFS (Ours)   |     0.77     |      0.68     |
> |    Oracle HL   |     0.81     |      0.75     |
>
> Robotic Manipulation, model-based
>
> | Representation |  O5  | O10 |
> |:--------------:|:----:|:---:|
> |    Raw Image   | 0.25 | 0.1 |
> |       VAE      |  0.6 | 0.3 |
> |       CPC      | 0.65 | 0.4 |
> |   VFS (Ours)   |   1  | 0.8 |
> |    Oracle HL   |   1  | 0.9 |
>
> This emphasizes that (1) VFS can work with imperfect skills and derive meaningful high-level policies, and (2) VFS enables learning a high-level policy that is close to an oracle HL baseline with most failure cases caused due to the skill failures.
>
>
>
> > Where does the goal state come from?
>
> That is correct -- it is the VFS encoded representation of a goal state/image provided to the high-level planner. We’ll update the paper with a clarification on this.
>
> > What are the skills available for Figure 6?
>
> This visualization was generated from the experiment setup described in Section 6.2, with access to the “MoveANearB” skills. We’ll update the figure and caption to clarify this better.
>
> > “=” sign in Figure 1(b)
>
> Thank you for the input, we did not realize the potential room for confusion in this regard. We’ll modify the figure to improve clarity.

---

> > ### Author Response · Authors · 2021-11-24
> > **Checking in**
> >
> > Dear Reviewer,
> >
> > We hope that you've had a chance to read our response. We would really appreciate a reply as to whether our response and clarifications have addressed the issues raised in your review, or whether there is anything else we can address.

---

> ### Author Response · Authors · 2021-11-27
> **Checking back**
>
> Dear Reviewer
>
> We hope that you've had a chance to read our response. We would really appreciate a reply as to whether our response and clarifications have addressed the issues raised in your review, or whether there is anything else we can address.

---

### Official Review · Reviewer_WuH2 · 2021-11-03

**Correctness:** 3
**Technical Novelty And Significance:** 3
**Empirical Novelty And Significance:** 3
**Recommendation:** 6
**Confidence:** 4

**Main Review:**

**Major comments**

*Source of options set:* The authors considered the options as pre-defined or learned using an options learning algorithm. However they assume that such a technique to construct the options also provides an associated value function for the options. However there are multiple options learning approaches in the prior research that do not provide an explicit value function for each of those options. Rather, the options are discovered through segmentation. The authors should provide a broader survey of options learning algorithms compatible with their approach and the popular approaches that are not compatible in the interest of better scholarship.

*Positioning with respect to prior work in options-based learning and planning:* The authors idea of using value functions of each of the pre-trained options in the state abstraction is not entirely novel. I can think of at least two works that have proposals along similar lines if not exactly the same. Please refer to the options keyboard by Baretto et al. [1] and AOSM by Rosen et al. [2]. I would like the authors to address their work in context of these works.

*Relationship with transfer learning and option reuse:* Works that attempt transfer learning through the reuse of options, are also themed around constructing better state features or abstractions tied to action availability and feasibility. The authors do not explicitly address this connection, and I would like to see them do that.

I quite like the idea of action abstractions, or action oriented constructions of state abstractions, especially for robotic domains, and this paper does propose a simple but potentially effective idea for achieving it. My primary concern centers around the fact that the presentation in the paper leave a lot to be desired in terms of positioning the paper with prior work, and tying it in with option discovery work that it relies upon.



[1] - Barreto, André, et al. "The option keyboard: Combining skills in reinforcement learning." NeurIPS(2019).

[2] - Rosen, Eric, et al. "Building Plannable Representations with Mixed Reality." 2020 IEEE/RSJ International Conference on Intelligent Robots and Systems (IROS). IEEE, 2020.

**Summary Of The Paper:**

This paper proposes that given a set of skills or options, the vector of the value functions for the individual skills becomes the abstract state representation in the approach proposed by this paper. The authors then go on to develop a model-free RL algorithm akin to DQN and a model-based planning algorithm for planning in the state-space defined by the value functions of the component skills, and the action-space being the set of executable options.

The authors demonstrate that the value function space serves as a better representation than some baseline algorithms for representation learning for RL in a maze solving and robotic manipulation task.

The key assumptions made are as follows:
 - The options are pre-defined and each option has an associated value function
 - The new task is solvable with the defined set of options.

**Summary Of The Review:**

I find value function spaces to be a simple but potentially effective idea. The comparisons have largely centered around representation learning frameworks applied to deep-RL, but not around reinforcement learning techniques aimed at option reuse and option discovery. While the approach proposed is different, these works deserve to be discussed in context. As such, I urge the authors to address these concerns qualitatively or empirically.

Post Response:

I am quite happy with the updated version of the paper, and willing to upgrade my score.

---

> ### Author Response · Authors · 2021-11-22
> **Response to Reviewer WuH2: updates to the writing and positioning**
>
> We thank the reviewer for their insightful comments and parallels to other approaches. We have amended our related work to discuss these works and place VFS in context. Changes in the main text are highlighted in red color. We are adding the following:
>
> > compatibility of options-learning frameworks with VFS
>
> We have *updated Section 3* with a broader survey of options learning algorithms compatible with our approach and popular approaches not directly compatible with our approach, but that can be coupled with policy evaluation to automatically obtain a value function -- this can be done via regression to empirical returns or temporal difference learning.
>
> > positioning wrt prior work in options-based learning
>
> We have *updated Section 2* with a discussion on the positioning of our work wrt prior works in options-based learning. We would like to thank the reviewer for pointing this out -- this is an important line of work we draw inspiration from and missed referencing it in the first draft.
>
> > relationship with transfer learning
>
> We have *updated Section 2* with a discussion on approaches that optimize for transfer learning to learn meaningful state abstractions.

---

> ### Author Response · Authors · 2021-11-27
> **Checking back**
>
> Dear Reviewer,
>
> We hope that you've had a chance to read our response. We would really appreciate a reply as to whether our response and clarifications have addressed the issues raised in your review, or whether there is anything else we can address.

---

### Official Review · Reviewer_aM2S · 2021-11-09

**Correctness:** 3
**Technical Novelty And Significance:** 2
**Empirical Novelty And Significance:** 3
**Recommendation:** 6
**Confidence:** 4

**Main Review:**

Strengths
-------------

- Well written paper: clear, to the point and nice to read
- VFS is a very interesting, simple and elegant state representation method
- VFS appears to work very well in practice (if the low-level skills are selected/trained by hand). I particularly liked the manipulation example: this is a challenging task and VFS seem to be quite effective

Weaknesses
------------------

- In the HRL setting, the most challenging task is discovering and identifying the low-level skills automatically. This is not touched at all by the paper. Rather the authors pick well-selected and fully trained low-level policies. I would have liked to see one of the following:
    - A setting where the low-level skills are not "perfect". A few examples consist: skills that overlap, skills that are not well-trained, skills that achieve the desired effect but in a sub-optimal way (e.g. make a big curve to reach a point), etc..
    - Experiments on how to integrate VFS inside a full HRL pipeline that automatically discovers low-level skills and uses them for planning with VFS representation; this is would actually be the use case of VFS (since it is assuming a rather prior-free setting).

Without one of the two it is actually difficult to assess if VFS is actually useful.
- "*in Figure 1b, states with varying object or arm positions (i-iii), different background textures (iv), and distractor objects (v) are functionally equivalent for planning, and map to the same embedding in our representation*" -> How is this true? Isn't $Z_t = [V_{o1}(s_t), ..., V_{ok}(s_t)]$? Passing the raw image state inside the VF learned by an RL algorithm will definitely affect the outcome and thus the high-level states will not necessarily be close to each-other. I'd like to see more discussion on this as this is quite crucial.
- Similarly, "*This representation captures positional information about the contents of the scene, preconditions for interactions, and the effects of executing a feasible skill, making it suitable for high-level planning*" -> Although I understand what the authors want to convey, this sentence is overselling what VFS do. Or at least there is no real validation in the paper suggesting that it does all of those things.
- By the way, especially in the MiniGrid world (that has a field of view of the agent as the raw observation) many (different) states will actually be encoded very similarly using VFS (see the Go-Explore paper {1} for a big discussion on this). How is the algorithm addressing this? What is the intuition behind the success in the MiniGrid world?
- The success rate metric is nice but not the only interesting metric. E.g. in the manipulation task, the algorithm might be able to position the objects but make unnecessarily big curves. At least a cumulative reward metric should also be defined (even if this is not given to the algorithms for learning).
- There is no mention of the neural network architectures for the VFs.

References
----------------

{1}: Ecoffet, A., Huizinga, J., Lehman, J., Stanley, K.O. and Clune, J., 2021. First return, then explore. Nature, 590(7847), pp.580-586.

**Summary Of The Paper:**

The paper presents a novel state representation technique that is based on Value Functions (VFs). The core idea of the paper is to use VFs to construct a high-level space representation in an hierarchical reinforcement learning (RL) scenario. The contributions of the paper are:

- Value Function Spaces (VFS): learned state representation
- Practical algorithms with VFS both for model-free and model-based RL settings
- Evaluation of VFS in 2 scenarios (MiniGrid and manipulation task)

**Summary Of The Review:**

The main idea of the paper is interesting and as far as I can tell novel. The results indicate that it is working well. However, the most difficult part in an hierarchical RL setting is actually discovering the low-level skills and how to identify them automatically. The paper is not touching this at all and assumes access to pre-defined (well-selected) low-level skills.

---

> ### Author Response · Authors · 2021-11-17
> **Response to Reviewer aM2S: imperfect skills, new experiments and qualitative results [1/2]**
>
> Thank you for the insightful feedback and comments. We appreciate the comments that you found the writing “clear and nice to read” and the idea “elegant” and “interesting”, as well that it works “very well in practice”. We reply to each comment below and are running additional tests to answer unanswered questions.
>
> > The most challenging task is discovering and identifying the low-level skills automatically… and Experiments on how to integrate VFS inside a full HRL pipeline...
>
> While we agree this is a very challenging and important problem for HRL, we argue that the problem of constructing skill-centric state abstractions from already existing skills is a challenging problem in its own right and find that using the value functions corresponding to these skills is a great way to obtain such abstractions “for free”. Discovering meaningful skills automatically, as studied extensively in the exploration and options literature is a very relevant next step to make VFS more feasible in an end-to-end approach and we hope that this paper provides a good foundation for future work in this direction, which we discuss in Section 7.
>
> > A setting where the low-level skills are not "perfect".
>
> We would like to clarify that while we assume access to pretrained policies, they are not necessarily “perfect”. In fact, our experiments had access to skills that were pretrained in a multi-task setting with about 60-85% task success in the robotic manipulation domain and an success-weighted path length of 0.75-0.9 in maze-solving (signifying imperfect skills with suboptimal behavior), and had significant overlap (for robotic manipulation, the MoveNear skills are heavily correlated e.g. skills like MoveCoffeeCupNearBowl shares attributes with MoveChocolateNearBowl or MoveBowlNearCoffeeCup, or GoToRedDoor and GoToGreenDoor in MiniGrid). To emphasize this further, we conducted some additional set of experiments using an oracle baseline: We use a human expert as HL planner to obtain this baseline. This denotes a “perfect” high-level policy. The failure cases with this baseline denote the cases where the imperfect low-level skills led to failure.
>
> Maze-solving, model-free HRL:
>
> | Representation | MultiRoom-10 | KeyCorridor-7 |
> |:--------------:|:------------:|:-------------:|
> |       AE       |     0.34     |      0.33     |
> |       CPC      |     0.37     |      0.35     |
> |       VAE      |     0.58     |      0.50     |
> |      CURL      |     0.43     |      0.54     |
> |   VFS (Ours)   |     0.77     |      0.68     |
> |    Oracle HL   |     0.81     |      0.75     |
>
> Robotic Manipulation, model-based:
>
> | Representation |  O5  | O10 |
> |:--------------:|:----:|:---:|
> |    Raw Image   | 0.25 | 0.1 |
> |       VAE      |  0.6 | 0.3 |
> |       CPC      | 0.65 | 0.4 |
> |   VFS (Ours)   |   1  | 0.8 |
> |    Oracle HL   |   1  | 0.9 |
>
> This emphasizes that (1) VFS can work with imperfect skills and derive meaningful high-level policies, and (2) VFS enables learning a high-level policy that is close to an oracle HL baseline with most failure cases caused due to the skill failures.
>
> > How are the learned representations in Fig 1b close to each other?
>
> VFS maps these states to similar states as they all have similar affordances. To construct the VFS the image is input to the value function for each task and each task outputs that state’s value. In a sparse reward setting, empirically we may expect this is something like 1 if the task is done, 0 if it is impossible, a high value if it is easy or achievable in a few timesteps, and a low value if it is hard or many timesteps away.
>
> Given skills trained with sufficient diversity (e.g. a skill taught to pick up a blue object from a clutter), we expect the learned values to be invariant to distractors that do not directly affect task performance (e.g. the presence of a background object). The VFS states being mapped to nearby locations is a by-product of the values being robust to such distractors. Similarly, the arm positions don’t matter much for the values of a drawer opening task as long as the arm is empty and able to open the drawer. We demonstrate this behavior more concretely in the robotic manipulation domain (Figure 6), where observations with different arm positions and background distractor objects are mapped to nearby embeddings by VFS.
>
> We are expanding discussion in the paper to make this clearer.

---

> > ### Author Response · Authors · 2021-11-17
> > **Response to Reviewer aM2S: imperfect skills, new experiments and qualitative results [2/2]**
> >
> > > “The representation captures preconditions and effects...” is not justified.
> > We thank the reviewer for pointing this out. To justify this qualitatively, we provide a few example rollouts of the robotic manipulation environment showing the evolution of the VFS state along the rollout. For each of the following figures, the x-axis is timestep, the y-axis is the value function, and the red skill is the selected skill being executed.
> >
> > 1. In this rollout, we notice that the only object on the table is a coffee cup and the corresponding value is high (capturing precondition). On picking up the cup, the value goes to 1, capturing the success outcome: https://imgur.com/a/LwAmgwq
> >
> > 2. In this rollout, there are three objects on the table (coffee cup, bottle and bowl) and the corresponding values are high (capturing preconditions). When the cup is picked up, the value corresponding to it shoots to 1 (capturing the success): https://imgur.com/a/Yi0ebfL
> >
> > 3. This rollout is slightly more complicated with multiple objects but if we focus on the bowl and apple that are present in the scene, we notice that the apple is inside the bowl. As the robot proceeds to pick up the bowl, the values corresponding to pick skills for both the bowl and action shoot up until the last time step -- the apple falls off the bowl and its value tanks. This is a complex affordance captured by the VFS state that would be otherwise very hard to represent using a learned state estimator: https://imgur.com/a/wHKrMTp
> >
> > While this was only a small number of examples that break down the rollouts with VFS, such behavior is expected of VFS as it represents the “functional information” of the observations and hence, qualitatively, captures the preconditions and outcomes.
> >
> > > How does VFS handle partial observability in minigrid environment?
> >
> > We use a modified version of MiniGrid that removes the partial observability aspect of using limited field of view egocentric agent observations by using a static top-down observation. We will add more details about this environment to alleviate the confusion, adding “in a fully observable setting, where the agent receives a full top-down view of the environment rather than an egocentric local image” to the main body and discussion in Appendix A. 1.
> >
> > > The success rate metric is not enough to capture task performance.
> >
> > That is a great suggestion! We have run our experiments again and evaluated the SPL for the minigrid environment below. We’re not sure of how to design an equivalent metric for the robotic manipulation domain, please let us know if you have any pointers.
> >
> > | Representation | MultiRoom-4 | MultiRoom-10 | KeyCorridor-3 | KeyCorridor-7 |
> > |:--------------:|-------------|:------------:|---------------|:-------------:|
> > |       Raw      |     0.42    |     0.23     |      0.41     |      0.23     |
> > |       AE       |     0.57    |     0.29     |      0.52     |      0.24     |
> > |       CPC      |     0.62    |     0.34     |      0.55     |      0.27     |
> > |       VAE      |     0.62    |     0.42     |      0.73     |      0.42     |
> > |      CURL      |     0.66    |     0.36     |      0.72     |      0.45     |
> > |   VFS (Ours)   |     0.84    |     0.68     |      0.72     |      0.59    |
> > |   Oracle HL   |     0.88    |     0.74     |      0.76     |      0.66     |
> >
> > We also added a new oracle baseline that does perfect planning at the high-level. We observe that the relative performance of the methods is largely unchanged, and the imperfect nature of skills causes a 10-20% reduction in SPL values across the board. When comparing performance to the oracular baseline, we notice that VFS achieves a performance close to the oracle, suggesting that the common failure cases are related to the imperfect skills and not due to planning.
> >
> > > There is no mention of the neural network architectures for the VFs.
> >
> > We are updating the submission (Appendix A) with details about the architectures for the skill VFs as well as the high-level policy. Here’s a summary of what we will be adding:
> > - Maze-Solving: We use a neural network with 4 convolutional layers followed by 2 FC layers as the estimator network inside a DQN/DDQN. For the high-level, we use 4 FC layers on top of the learned representations.
> > - Robotic Manipulation: We use a MobileNetv2 encoder followed by 2 FC layers for the estimator network. For the high-level, we use 3 FC layers on top of the learned representations.

---

### Decision · Program_Chairs · 2022-01-20

**Decision:**

Accept (Poster)

**Comment:**

This paper is good but at a borderline. One reviewer increased the score during the discussions. However, no reviewer was in strong favor. So that this paper is still a borderline one, and it is up to the SAC to decide.